# The Distribution of Surface Soil Moisture over Space and Time in Eastern Taylor Valley, Antarctica

Mark R. Salvatore [1,*], John E. Barrett [2], Laura E. Fackrell [1,3], Eric R. Sokol [4], Joseph S. Levy [5], Lily C. Kuentz [5,6], Michael N. Gooseff [7], Byron J. Adams [8], Sarah N. Power [2], J. Paul Knightly [1], Haley M. Matul [1], Brian Szutu [1] and Peter T. Doran [9]

1   Department of Astronomy and Planetary Science, Northern Arizona University, Flagstaff, AZ 86011, USA
2   Department of Biological Sciences, Virginia Polytechnic Institute and State University, Blacksburg, VA 24060, USA
3   Biotechnology and Planetary Protection Group, NASA Jet Propulsion Laboratory, California Institute of Technology, Pasadena, CA 91109, USA
4   National Ecological Observatory Network, Battelle, Boulder, CO 80301, USA
5   Department of Earth and Environmental Geoscience, Colgate University, Hamilton, NY 13346, USA
6   Department of Geography, University of Oregon, Eugene, OR 97403, USA
7   Institute of Arctic and Alpine Research, University of Colorado Boulder, Boulder, CO 80309, USA
8   Department of Biology, Brigham Young University, Provo, UT 84602, USA
9   Department of Geology and Geophysics, Louisiana State University, Baton Rouge, LA 70803, USA
*   Correspondence: mark.salvatore@nau.edu; Tel.: +1-(928)-523-0324

**Abstract:** Available soil moisture is thought to be the limiting factor for most ecosystem processes in the cold polar desert of the McMurdo Dry Valleys (MDVs) of Antarctica. Previous studies have shown that microfauna throughout the MDVs are capable of biological activity when sufficient soil moisture is available (~2–10% gravimetric water content), but few studies have attempted to quantify the distribution, abundance, and frequency of soil moisture on scales beyond that of traditional field work or local field investigations. In this study, we present our work to quantify the soil moisture content of soils throughout the Fryxell basin using multispectral satellite remote sensing techniques. Our efforts demonstrate that ecologically relevant abundances of liquid water are common across the landscape throughout the austral summer. On average, the Fryxell basin of Taylor Valley is modeled as containing $1.5 \pm 0.5$% gravimetric water content (GWC) across its non-fluvial landscape with ~23% of the landscape experiencing an average GWC > 2% throughout the study period, which is the observed limit of soil nematode activity. These results indicate that liquid water in the soils of the MDVs may be more abundant than previously thought, and that the distribution and availability of liquid water is dependent on both soil properties and the distribution of water sources. These results can also help to identify ecological hotspots in the harsh polar Antarctic environment and serve as a baseline for detecting future changes in the soil hydrological regime.

**Keywords:** hydrology; soil moisture; remote sensing; dry valleys; ecology

## 1. Introduction

Freshwater availability is the most critical environmental variable influencing ecosystem function in the McMurdo Dry Valleys (MDVs) of Antarctica [1–4]. The presence of liquid water implies surface temperatures (soil and/or air) above freezing (noting that solutes may depress freezing points below 0 °C), which almost exclusively occur during the summer months when the region is illuminated by 24 h of sunlight (topography and shadows permitting). When liquid water is absent, the multitude of Antarctic cyanobacteria, nematodes, rotifers, and tardigrades undergo a variety of physiological survival mechanisms, including crypto-protective dehydration and the production of recrystallization-inhibiting and antifreeze proteins [5,6]. However, upon the reintroduction of liquid water, these

organisms have been shown to rapidly recover and can begin metabolizing within minutes, even after years of anhydrobiosis [7].

The distribution and structure of ecological communities in the MDVs are both spatially and temporally heterogeneous due to the extreme spatial and temporal heterogeneity in the physical controls and environmental conditions that shape this desert ecosystem [8–11]. Specifically, previous studies have investigated how spatial and temporal variations in soil moisture availability influence the composition and function of MDV soil ecosystems. For example, [12] found that soil moisture content was the only property related to the diversity of microeukaryotes in the soils of the MDVs. They also found that biodiversity significantly increases at soil moisture contents greater than 3%. Ref. [5] described how the moisture content of soils can influence the rate of temperature change during the transition towards freezing temperatures, which, in turn, influences the preferred physiological strategies implemented to cope with freeze-induced stress [13,14]. The differences in physiological mechanisms adapted to coping with dehydration and freezing likely explain the key differences in community structure as well; for example, nematode communities dominated by *S. lindsayae* typically occur in drier soils because of their inability to effectively dehydrate or resist freezing when high soil moisture contents are present [5,15]. Alternatively, the *P. murrayi* and *E. antarcticus* species of nematodes are more commonly found in wetter soils because of their range of freeze-tolerance strategies that can result in slow dehydration or the creation of antifreeze and cold-resistant proteins prior to complete freezing [16].

Ref. [15] studied the coiling of soil nematodes as an indicator of anhydrobiosis and inactivity in MDV soils of varying soil moisture content. They found that nematode coiling reached a maximum of ~80% in soils with <2% gravimetric water content (GWC; mass of water per mass of dry soil). Nematode coiling consistently decreased in soils with GWC between 2% and 12%, and nearly all nematodes were uncoiled within six hours of reaching 12% GWC [15]. Ref. [7] found that reactivating an abandoned stream channel after nearly two decades of inactivity led to the quick reactivation of cyanobacterial communities within the channel "without significant lag after the onset of restored stream flow", and Ref. [17] showed that areas of regularly enhanced soil moisture are characterized by significantly different geochemistry and host distinct biotic communities compared to the surrounding soils. These (and many other) studies demonstrate how local ecosystem structure and function are largely dependent on the availability of liquid water and its activity and persistence across the landscape.

To summarize, previous studies have shown that both biodiversity and biological activity in the surface soil environment increase with increasing soil moisture, with a critical threshold between regions of high and low present at around ~3% GWC. So where are these areas of high and low biodiversity and metabolic activities located? Identifying all of these areas using field observations alone is impossible due to the intensity of the required measurements and the logistical challenges of Antarctic field work (e.g., access and transportation).

The ability to remotely determine soil moisture over both space and time has been long sought after in the MDVs, where soil conditions can change rapidly over timescales of minutes and where global climate warming is changing the baseline of soil conditions. Most satellites specifically designed to study soil moisture are not well suited for this type and scale of analysis. For example, the ~40 km spatial resolution of the Soil Moisture Active Passive (SMAP) mission is approximately four orders of magnitude greater than the scale of ecosystem processes occurring in the MDVs [18]. Additionally, the penetration depth of C-band radar such as that employed on the RADARSAT-2 mission varies greatly as a function of soil moisture itself, ranging from less than 1 mm depth at ~35% volumetric water content to more than 10 cm at <5% volumetric water content [19]. Therefore, the use of high-resolution visible/near-infrared (VNIR) reflectance spectroscopy and visible imaging has been the most utilized method for studying soil moisture in the MDVs over the last decade. Ref. [20] demonstrated how the flow of water in the shallow subsurface of

the MDVs can be identified from orbit and how their flow paths correlate with topographic depressions and poorly channelized surface flow paths, making them predictable using remote sensing data. Then, [21] demonstrated how hyperspectral visible/near-infrared (VNIR) remote sensing data can be used to quantitatively predict GWC using the depth of the 1.4 μm absorption feature associated with $OH^-$ stretching vibrations and combination tones of the $H_2O$ molecule [22]. Lastly, refs. [23,24] demonstrated a novel technique for confidently detecting regions of enhanced soil moisture by quantifying observed soil darkening while also accounting for variations in surface shadows, illumination, and the underlying geological properties of the surface. These techniques are essential for using remote sensing and spectroscopic data to confidently detect areas of increased soil moisture in the absence of direct field measurements.

In this study, we derive the spatial and temporal distribution of surface soil moisture using high-resolution multispectral VNIR remote sensing data. We build upon the work of [23] to quantify the relationship between observed surface albedo and surface soil moisture content. We also use both a high-resolution lidar-derived digital elevation model (DEM) to remove variable shadows caused by different solar illumination geometries as well as laboratory experiments using field-collected soil and sediment. We focus exclusively on the area surrounding Lake Fryxell (termed the Fryxell basin) in Taylor Valley, Antarctica, because of the availability of remote sensing data and the density of stream channels, water tracks, and diverse microbial and invertebrate communities.

## 2. Background

### 2.1. Soil Moisture in the McMurdo Dry Valleys

The MDVs of Antarctica consist of several east–west trending glacial valleys carved perpendicular to the spine of the Transantarctic Mountains (TAM). Bound to the west by the East Antarctic Ice Sheet (EAIS) and to the east by the Ross Sea, the MDVs remain free of permanent glacial ice due to the buttressing of the EAIS by the TAM and minimal snow accumulation due, in part, to the strong, warm, dry, gravitationally driven foehn winds [25]. Taylor Valley is one of these glacially carved valleys that is largely free of perennial snow and ice cover from the toe of Taylor Glacier in the west to the Ross Sea in the east. Three large basins hosting perennial frozen lakes are present in Taylor Valley, with Lake Bonney adjacent to the toe of Taylor Glacier in the west, followed by Lake Hoare, and finally Lake Fryxell furthest to the east and closest to the Ross Sea (Figure 1). The valley widens and flattens from west to east, with Lake Bonney and Lake Hoare adjacent to steep valley walls and Lake Fryxell flanked by relatively shallow slopes. While other valleys also host permanently frozen glacial lakes (e.g., Lake Vida in Victoria Valley, Lake Vanda in Wright Valley), Taylor Valley is unique in the number of large lakes and both the abundance and density of glacial melt streams that supply significant quantities of liquid water to the lakes of Taylor Valley.

The meteorology of the MDVs is strongly seasonal due to their polar setting, with continuous daylight during the ~3 months of summer, continuous darkness during the ~3 months of winter, and vernal and autumnal transitions between the two endmembers. The mean annual temperature in the Fryxell basin is −20.0 °C, with mean summer and winter temperatures recorded at −7.7 °C and −30.2 °C, respectively [26,27]. The mean annual degree days above freezing increase with distance from the Ross Sea, ranging from 13.1 days at Explorer's Cove, to 20.9 days at Lake Fryxell, to 44.2 days at Lake Bonney [27]. Precipitation is almost exclusively in the form of snow with 3–50 mm of water equivalent annual precipitation, the majority of which sublimates (with minimal melting) before it is able to accumulate for extended periods [28]. While the mean annual soil temperatures in the Fryxell basin average approximately −19 °C down to a depth of 10 cm, maximum recorded soil temperatures at 0 cm, 5 cm, and 10 cm depths have been recorded at +24.9 °C, +14.7 °C, and +12.0 °C, respectively, demonstrating how the soils have the potential to serve as a refugia for microbial and invertebrate communities [27].

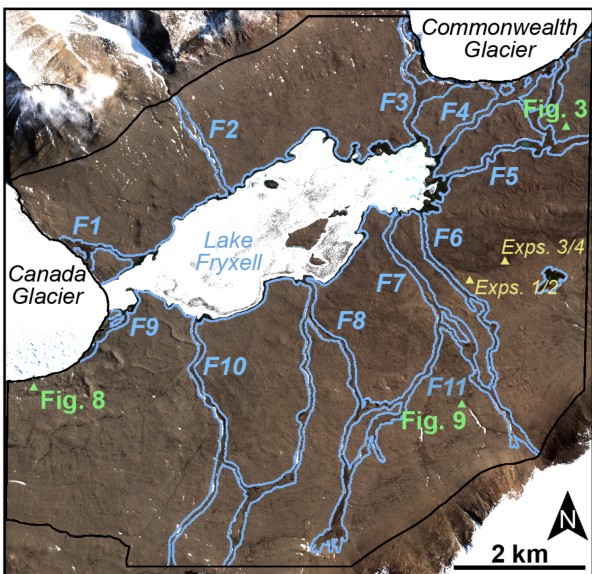

**Figure 1.** Major hydrologic features of the Fryxell basin, eastern Taylor Valley, Antarctica. Streams are labeled according to their USGS designated numbers, which correspond to the following: (F1) Canada Stream; (F2) Huey Creek; (F3) Lost Seal Stream; (F4) McKnight Creek; (F5) Aiken Creek; (F6) Von Guerard Stream; (F7) Harnish Creek; (F8) Crescent Stream; (F9) Green Creek; (F10) Delta Stream; and (F11) the relict channel. The locations of other figures are marked with green triangles, and the locations where sediments were collected for our experiments are indicated with yellow triangles. Imagery © 2019 Maxar.

The surface geology of the Fryxell basin is dominated by the Ross Sea drift, a glacial mix of granitic, metamorphic, and basaltic till sourced from a grounded ice lobe from the western Ross Sea and extending into the mouth of Taylor Valley [29]. Local compositional variations can be observed both from the ground and from orbit and demonstrate the extent of heterogeneity observed over a range of spatial scales. Grounded ice is thought to have diverted into Taylor Valley as it moved to the north in the Ross Sea, as inferred from the composition of glacial erratics [29,30]. The grounded ice reached a maximum elevation of ~320 m above sea level in Taylor Valley and dammed meltwater from Taylor Glacier behind it, creating glacial Lake Washburn that occupied the majority of the Fryxell basin from ~11–24 ka before present [29–33]. The Fryxell basin is also geochemically distinct from the more inland basins of Taylor Valley. The basin contains less than 10% of the total salt content of the Bonney basin, and those salts are dominated by $Na^+$ and $HCO_3^-$ relative to the $Ca^{2+}$, $Cl^-$, and $SO_4^{2-}$ salts that dominate the Bonney basin [32]. In addition to the compositional influences of the Ross Sea drift, mounds, kettle ponds, and lateral and longitudinal ridges occupy the entirety of the Fryxell basin, creating a complex morphological landscape on which the modern hydrological and ecological systems are situated.

Soils developed on glacial sediments have been thoroughly described in the MDVs by previous investigators (e.g., [34,35]). All soils in the Fryxell basin are characterized by a surface lag deposit of coarse sands, granules, and pebbles and are underlain by more sand-rich subsurface horizons [36,37] with fine fractions (fine sand, silt, and clay) increasing with proximity to the Ross Sea coast [38]. While the dominance of coarse materials in the upper ~5 cm is due primarily to winnowing, deflation, and aeolian redistribution, the subsurface structure of MDV soils suggests that sedimentary processes (dominated by physical erosion and other periglacial processes) play more of a role in MDV soil development and vertical structure than winnowing alone [35,36]. Surface lags and desert pavements generally increase in particle size with age as coarse-grained clasts continue to break down into smaller particle sizes, leaving predominantly ventifacted fine-grained clasts at the surface [35]. The relatively low density of large clasts or boulders as well as the prevalence of angular granitic granules and pebbles suggest that the soils of the

Fryxell basin are young and poorly developed compared to other surfaces throughout the MDVs [35]. The coarse nature of most Taylor Valley soils produces well-drained soils with large pore spaces, creating a compromise between high permeability (which promotes water migration) and low capillarity (which prevents the wicking of water due to surface tension and is controlled by the sample's matric potential) [39,40].

Glacial melt streams are active for approximately 4–8 weeks per year [5] when summer temperatures are warmest and daily solar insolation incident upon the glaciers is greatest. The adjacent hyporheic zones of interacting surface waters, shallow groundwaters, and saturated sediments act as critical buffers as they exchange water with the main channels [41]. At this time, the soils of the MDVs also become "active" and can become saturated through the localized melting of shallow permafrost, shallow snowmelt-fed non-channelized flow along the ice table (termed water tracks), and/or localized deliquescence of water from the atmosphere into the soils [20,42,43]. All active stream channels and water tracks act as "superhighways" for transporting water, dissolved ions, and dissolved organic matter from the surrounding landscape to the Lake Fryxell lacustrine system [20,44]. More generally, and in places where channelized flow is not observed to occur, liquid water can still be produced by several processes (Figure 2), as follows:

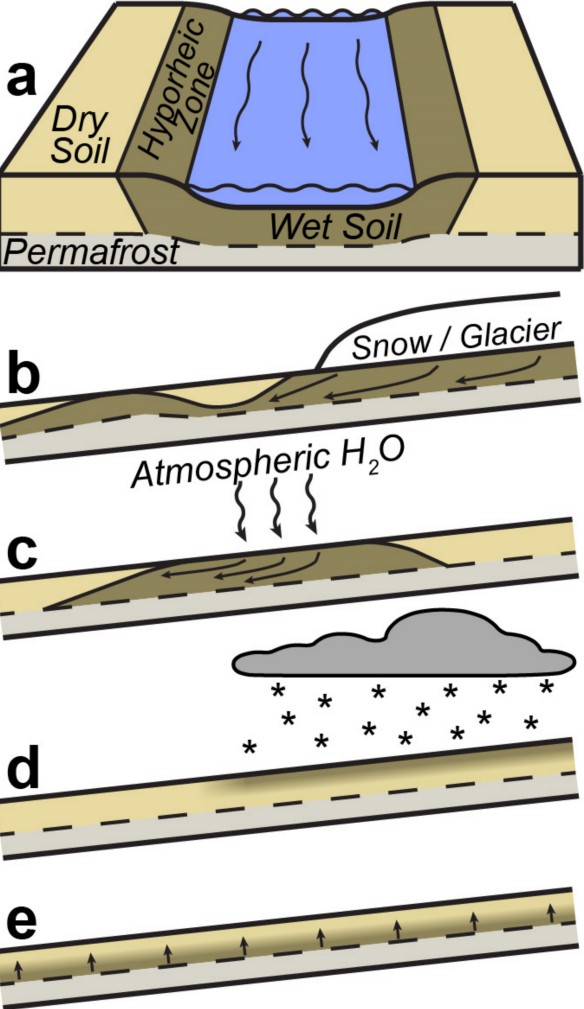

**Figure 2.** Primary sources of soil moisture in the McMurdo Dry Valleys of Antarctica: (**a**) wetting of the hyporheic zone by nearby overland flow; (**b**) soil moisture sourced from the melting of uphill snow or glaciers; (**c**) atmospheric deposition of moisture (deliquescence); (**d**) direct precipitation; and (**e**) capillary wicking from melting of buried ground ice.

- **Migration of water due to surface or subsurface flow.** Soils adjacent to stream channels or lake margins receive a regular supply of liquid water as a result of surface or subsurface flow [41]. These are some of the few locations in the MDVs where surface soils can remain saturated throughout the active season.
- **Melting of surface snow or subsurface ice deposits.** The surface snow and buried ice deposits that do form throughout the valleys (typically in topographically sheltered locations where windblown snow can accumulate) are subject to continual melting throughout the active season, creating small refugia throughout the landscape that can support processes that demand access to more regular sources of liquid water [45,46]. Relatedly, the majority of the MDVs are underlain by shallow ($\lesssim$1 m depth) ground ice, which has been shown to be subject to melting as a result of both a warming climate and due to unique landscape evolution processes [47,48].
- **Deliquescence.** The presence of abundant hygroscopic salts in the soils of Taylor Valley can result in the direct absorption of atmospheric water vapor when relative humidity exceeds a salt-dependent critical threshold [49]. Both laboratory [43,50,51] and field observations [51–53] have demonstrated the local effects of salt deliquescence in the MDVs.
- **Direct precipitation.** Snowfall is common throughout the MDVs [54]. Snow usually falls in small quantities and succumbs primarily to sublimation before melting can occur [28]. However, some snow is able to melt and briefly increases surface soil moisture content.

### 2.2. Relating Soil Moisture and Albedo

Studies relating VNIR spectral signatures and soil moisture content have been conducted since the early days of remote sensing. For example, Ref. [55] performed a series of laboratory experiments comparing soil moisture content (among other soil properties) to the spectral responses at Landsat Thematic Mapper wavelengths. They also highlighted potential issues associated with surface topography and variations in solar illumination. Since these early studies, additional efforts have characterized and refined models to more accurately determine the response of soil reflectance to a range of soil properties (e.g., [56–58]). Despite the numerous encountered challenges [58], hundreds of published investigations have demonstrated the utility of using remote sensing data to characterize variations in soil moisture content in environments ranging from tropical (e.g., [59]) to the Antarctic (e.g., [21,53]).

Several mechanisms have been proposed to explain the relationship between soil albedo and soil moisture content [60]. Ref. [61] first deduced that the darkening of soils with increased soil moisture was the result of increased scattering of light through thin films of water before and after interacting with individual soil particles. They and others (e.g., [62]) also suggested that internal reflection, refraction, and scattering of light in interstitial soil water would effectively "trap" radiation and lead to greater extinction. Ref. [61] proposed the following relationship between "wet" and "dry" soils using first principles:

$$r_{wet} = \frac{r_{dry}}{\left[ n^2 \left( 100 - r_{dry} \right) + r_{dry} \right]} \times 100\% \tag{1}$$

where $r_{wet}$ and $r_{dry}$ are the reflectance values (in percent) of wet and dry soils, respectively, and $n$ is the index of refraction for liquid water [60,61]. Instead of a direct relationship with water content, Ref. [63] proposed that the primary control on soil albedo is not water content, but instead water potential, which varies as a function of particle size and other soil properties. This mechanism would more directly explain why different soils reach minimum observed albedos at different soil moisture contents. However, Ref. [60] demonstrated that the water potential at which maximum darkening occurs is not the same for the different soils measured in their study, suggesting instead that finer soils reach maximum darkening at lower water potentials than coarser soils. They justifiably

conclude that the relationship between surface albedo and soil moisture content is a complex interplay between many different soil properties.

Nearly all studies relating spectral signatures with soil moisture identify at least two regimes in their data (e.g., [60,64–67]): one where soil moisture and observed albedo show a strong negative correlation at relatively low soil moisture contents, and another where such a relationship breaks down at relatively high soil moisture contents [60,61,64,67]. We refer to these different regimes as "optically unsaturated" and "optically saturated", respectively. When a soil is optically unsaturated, increasing soil moisture content will lead to a fairly predictable decrease in the overall soil albedo. However, studies have shown there to be a "critical value" of soil moisture where the effects of increasing soil moisture no longer have a predictable influence on the observed albedo (e.g., [67]). When soil moisture contents are in the optically saturated regime, it is impossible to quantitatively relate these two properties [64]. Others (e.g., [66]) have suggested that the relationship between soil moisture content and surface albedo is an exponential decay function, although the nature of this relationship is closely related to particle size and other soil properties [64].

## 3. Methods

Calculating the GWC of Antarctic soils using remote sensing data is rooted in the assumption that observed surface darkening is due to increased soil moisture. While the absence of vascular plants resolves one possible challenge to this assumption, differences in solar illumination and viewing geometries also result in perceived surface brightening and darkening that must be addressed. We have developed a combined remote sensing and laboratory analysis workflow to: (1) calibrate high-resolution satellite data to quantitative surface brightness (albedo), (2) correct each pixel for the influence of solar illumination and topography, (3) compare the corrected albedo at each pixel to a long-term baseline surface albedo product to determine the amount of darkening or brightening observed at each pixel, and (4) derive GWC at each pixel through comparison to laboratory measurements.

### 3.1. Remote Sensing Image Calibration and Topographic Correction

We identified 57 high-quality (i.e., minimal clouds or surface snow cover) 8-band WorldView-2 and -3 (WV02 and WV03) images collected over the Fryxell basin between December 2009 and December 2019 (Table A1). The images were calibrated using the methods of [23], where data were first calibrated to top-of-atmosphere (TOA) reflectance and then quantitatively compared to five invariant ground validation sites to derive and remove the estimated atmospheric contributions from each image. Band-specific regressions to correlate satellite-derived TOA reflectance with field-derived surface reflectance were calculated and applied to each pixel, resulting in calibrated and validated surface reflectance WV02 and WV03 data. All eight spectral bands for each pixel were then averaged to derive the mean VNIR surface albedo at each pixel.

Like all VNIR satellite data, the dominant control on surface brightness is typically associated with variations in shading and illumination due to local topography (Figure 3a). We used a high-resolution digital elevation model (DEM) derived during an airborne lidar survey collected during the 2014/2015 austral summer [68] to model the influence of topography and solar illumination for each of the 57 WV02 and WV03 images collected over the Fryxell basin. The solar elevation and azimuth were obtained from each image's corresponding metadata product and were used to generate a surface hillshade product that is illuminated in a manner identical to each satellite image (Figure 3b). Quantitative relationships between VNIR surface albedo (generated above) and the relative albedo of each pixel in the hillshade products were derived using linear regressions and were used to produce illumination-predicted surface albedo images. These products remove topographic influences and allow for images acquired under many different viewing and illumination geometries to be compared simultaneously. We then divided the derived surface albedo data from the 57 individual WV02 and WV03 images by their respective lidar- and illumination-predicted surface albedo images to generate normalized surface

albedo products, where all values are normalized to the illumination-predicted surface albedo data products (Figure 3c).

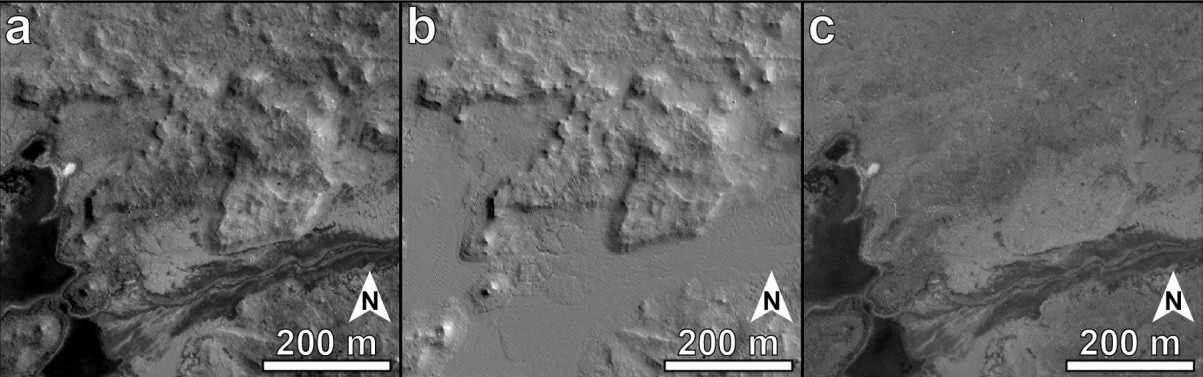

**Figure 3.** Demonstrating the removal of topographic shading from high-resolution satellite images: (**a**) a panchromatic WorldView-2 image of the Many Glaciers Pond region of eastern Taylor Valley, Antarctica; (**b**) a topographic hillshade generated using 1 m lidar topographic data [68] and solar geometry information obtained from the WorldView-2 image metadata file; (**c**) the topographically removed WorldView-2 image created by normalizing the data to the hillshade-predicted albedo. Imagery © 2018 Maxar.

However, these normalized surface albedo products do not account for the inherent differences in surface brightness caused by different surface compositions or physical properties (e.g., soil type, clast composition, mineralogy). This is because the data used to derive the illumination-predicted surface albedo products only consisted of lidar-derived topographic data that are agnostic to any properties except solar illumination geometry. To account for the inherent surface albedo that is invariant over the timescales of our investigation, we combined all 57 normalized surface albedo products and generated a median normalized surface albedo product, which provided the median surface albedo for each pixel in all of the 57 images. Median values were used over average values to minimize the effect of short-term darkening (due to increased soil moisture) or brightening (due to salt precipitation or the presence of snow).

Finally, we selected the highest quality 21 WV02 and WV03 images in our database and divided the median normalized surface albedo product from each pixel in these 21 images. Dividing the median normalized surface albedo product from each image results in 21 new temporally constrained relative surface albedo products. Most pixels in these relative surface albedo products fall near a value of 1.0, indicating that the observed albedo at that pixel is consistent with the long-term baseline median albedo for that given location. Values greater than one indicate a temporary brightening (i.e., snow or salt) of the surface relative to the long-term baseline albedo, while values less than one indicate a temporary darkening (i.e., wetting).

The orbital data were assembled and processed using the ArcMap geographic information system (GIS) (ESRI, Redlands, CA, USA) and the Environment for Visualizing Images (ENVI) software (Harris Geospatial, Broomfield, CO, USA). To focus exclusively on the soils outside of the established stream channels, ponds, and lakes, we created a 20 m buffer on both sides of the thalwegs of each established stream as well as the margin of Lake Fryxell and other perennial bodies of water. Lastly, the regions of the Fryxell basin whose median surface albedo value fell far higher than neighboring pixels were omitted from our investigation altogether, as these high albedo values indicate persistent snowpacks that are present for the majority of the decade of orbital imaging.

### 3.2. Quantitatively Relating Relative Albedo to Gravimetric Water Content

To quantify the relationship between relative surface darkening and soil GWC, we conducted four lab experiments on two sets of soils from the Fryxell basin (Table 1). The

first set of soils was collected from a "typical" soil surface approximately 100 m east of Von Guerard Stream and ~1.2 km upslope from Lake Fryxell. The second set of soils was collected from ~700 m east of the first set of soils and within a dark basalt-rich ridge of the Ross Sea Drift [29]. These two sets of soil are hereafter referred to as the "typical" and "dark" soil sets, respectively. Approximately 3 kg of soils and surface clasts were collected from the upper ~5 cm of the soil column as a means of capturing the properties of the uppermost surface. Two sample aliquots were extracted from each set of soils: the first aliquot was sieved to particle sizes less than 4 mm in diameter (to characterize the granule- and pebble-rich surface lag that is ubiquitous throughout the Fryxell basin), while the second aliquot was sieved to particle sizes less than 1 mm in diameter (to characterize the behavior of the finer fraction that is not influenced by surface pebbles and clasts). In total, four experiments were run on four different sample sets/aliquots: Exp. #1 consisted of typical soils < 4 mm in diameter, Exp. #2 consisted of typical soils < 1 mm in diameter, Exp. #3 consisted of dark soils < 4 mm in diameter, and Exp. #4 consisted of dark soils < 1 mm in diameter.

The density of each experimental soil sample was measured using standard volumetric displacement methods where a known mass of sediment is placed into a graduated cylinder with a known volume of distilled water. After agitation, the volume of the sediment + distilled water and the known mass of dry sediment were used to calculate the bulk densities of the samples. Bulk hydrological properties were also derived for both < 4 mm size fraction (Exps. #1 and #3). Known masses of both samples were saturated with distilled water and allowed to gravitationally drain for 20 min before being reweighed. The ratio of this gravitationally drained to dry sediment mass is known as its field capacity, which is generally thought to be the upper bound of available soil moisture for the purposes of plant growth after all large pores are allowed to drain [69]. We also estimated the wilting point of our soil samples, which is formally defined as the GWC at $-1500$ kPa suction pressure and represents the minimum available soil moisture for plants to carry out biological functions [69]. We used the common convention of estimating the wilting point as 50% of the measured field capacity, which is consistent with prior studies that investigated soils with very low organic carbon content [70].

To determine the spectral properties of each soil sample as a function of soil moisture content, 5 cm$^3$ of each sample was subset into small weighboats and agitated to mimic gravitational settling as a result of aeolian redistribution [71]. Distilled water was added along the edge of the weighboats so as not to significantly disrupt the sample surfaces. Water was added to the weigh boats until (1) no additional darkening was observed, and (2) water was observed to be clinging to the uppermost grains, indicating that capillary suction was able to connect the surface and subsurface [69]. The samples were then placed on a balance (0.001 g precision) and moved under a full spectrum halogen lamp (located ~20 cm above the sample at 30° off-nadir) to both illuminate the sample for VNIR spectral characterization and to hasten the evaporation of water from the sample.

We used a high-resolution VNIR FieldSpec4 spectroradiometer (Analytical Spectral Devices, now Malvern Panalytical, Boulder, CO, USA) to assess the relationship between GWC and observed surface albedo. The fiberoptic receiver from the FieldSpec4 was positioned ~4 cm above the sample at nadir, equating to a spot size of roughly 2.5 cm$^2$. The spectroradiometer was calibrated to reflectance using standard dark current and white reference (Spectralon) calibrations. Spectra were acquired and sample weights recorded every ~60 s as the samples dried from above under the halogen lamp. Measurements were recorded until the continuously measured sample weight stabilized, indicating the complete removal of free water from the sample. The coordinated sample weights and spectral measurements were then used to calculate the spectral responses of surface sediments as a function of GWC. GWC was calculated by dividing the mass of water in the sample by the mass of the dry sample. All spectral data were downsampled to WV02 and WV03 resolutions and averaged to generate mean surface albedo for the direct comparison to orbital data.

**Table 1.** Properties of the four sediment samples investigated in this study.

| Source Location | Exp. # | Classification | Density (g cm⁻¹) | Measured Field Capacity/Modeled Wilting Point ᵃ (GWC) | Exp. Duration | No. of Collected Spectra | Albedo (Air Dry) |
|---|---|---|---|---|---|---|---|
| 77.61726°S, 163.28593°E (Typical Soils) | 1 | Very Coarse Sand | 1.62 | 10.6%/5.3% | 122 min | 112 | 0.173 |
| | 2 | Coarse Sand | 1.80 | | 58 min | 64 | 0.192 |
| 77.61418°S, 163.30864°E (Dark Soils) | 3 | Fine Sand | 2.06 | 12.3%/6.1% | 116 min | 117 | 0.159 |
| | 4 | Loamy Sand | 2.79 | | 77 min | 74 | 0.138 |

ᵃ Wilting point estimated as 50% of the measured field capacity [68].

### 3.3. Application to Remote Sensing Datasets

After generating an empirical relationship between relative VNIR albedo and GWC through our lab experiments (see Section 4), we applied this model to the 21 high-quality WV02 and WV03 images (Appendix A) to predict the GWC at each pixel in each image. We also identified the pixels in each image that fell above both the measured field capacity and the modeled wilting points described in Table 1. We chose to use the values associated with the dark soil samples; although these soils are not as compositionally representative of typical soils throughout the Fryxell basin, the higher GWC values associated with both the field capacity and wilting points will ensure that our model results are underestimates of soil moisture content as opposed to overestimates. We believe that the most valuable data products generated from these remote sensing data are: (1) the average GWC per pixel, (2) the standard deviation of measured GWC over time, (3) the fraction of pixels observed above the wilting point (GWC > 6.1%), and (4) the fraction of pixels observed above field capacity (GWC > 12.3%).

## 4. Results

### 4.1. Characteristics of Soil Samples

All four of our experimental soil samples can be classified as different varieties of sand, from very coarse sand (Exp. #1) to loamy sand (Exp. #4) (Figure 4 and Table 1). The typical soils were much coarser than the dark soils, with 19.6% of the bulk typical soil sample (Exp. #1) having particle sizes less than 250 μm and 77.8% of the bulk dark soil sample (Exp. #3) having particle sizes less than 250 μm. The relatively large particle sizes associated with the typical soil set results in a relatively gradual transition from sand-, to granule-, to pebble-sized particles. However, the finer nature of the dark soil samples does not preclude the surface from still being dominated by granule- and pebble-sized particles, which can be seen as an enrichment of these larger particles despite the lack of particles between 1 mm and 250 μm in size (Figure 4).

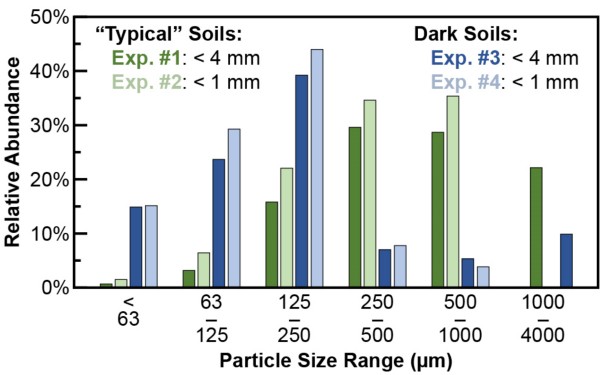

**Figure 4.** Particle size distributions for the four soil samples studied in this investigation.

Dark soils also showed a significantly greater bulk density than the typical soils, with the bulk soils (<4 mm particle sizes) having a 27% greater density and the finer soils (<1 mm particle sizes) having a 55% greater density than the more typical soils. Despite these differences in particle size distributions and densities, the measured field capacity of the bulk soils (Exps. #1 and #3) were found to be 10.6% and 12.3% GWC, respectively. Using the standard convention of estimating the wilting point as 50% of the GWC measured at field capacity, the estimated wilting points of Exps. #1 and #3 are 5.3% and 6.1%, respectively.

Prior to each set of measurements, the soil samples were manually agitated to mimic the influence of the aeolian redistribution and gravitational settling that are known to dominate the mobile soil component throughout the MDVs [35,36,72]. All four experimental soil aliquots exhibit comparable spectral signatures (Figure 5), with increases in reflectance through the visible wavelength range and several narrow vibrational absorption features centered near 1.4 μm ($OH^-$ stretching overtones and $H_2O$ combination tones), 1.9 μm ($H_2O$ bending and stretching combination tones), and between 2.1 and 2.5 μm (metal-OH stretching and bending combination tones) [22,73,74]. These narrow vibrational absorptions are ubiquitous throughout the MDVs and are the result of both amorphous and crystalline hydrated phases within the soils [75]. Shallow and broad absorptions centered near 1 μm are consistent with the presence of both $Fe^{2+}$ and $Fe^{3+}$ in crystal lattices, resulting in electronic charge-transfer and crystal field absorptions [22,76]. Additionally, the flattening of the spectra (where the reflectance linearly decreases) between 0.6 μm and 0.9 μm in all experiments (with the exception of Exp. #2) is likely the result of underdeveloped $Fe^{3+}$ crystal field absorptions [76]. Exp. #2 (<1 mm fraction of typical soils) exhibited weaker absorptions associated with the presence of $Fe^{3+}$, likely because the sample is bright and coarse grained, suggesting a lack of Fe-rich mafic minerals relative to the coarse surface fraction or the dark soil samples [29].

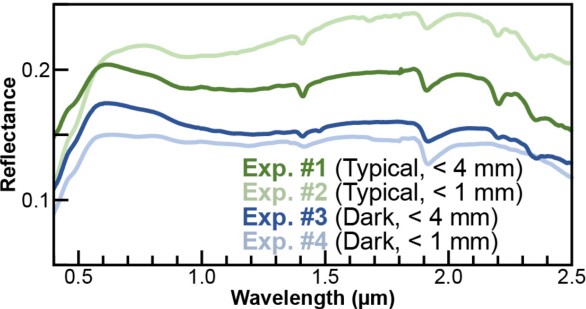

**Figure 5.** Visible/near-infrared reflectance spectra of the four soil samples studied in this investigation.

Exp. #2 was found to be 1.9% brighter than its corresponding bulk sample (Exp. #1), suggesting that the larger particles in the bulk sample act to lower the overall albedo of typical soil samples (Figure 5). In contrast, the finer fraction of the dark soils (Exp. #4) was found to be 2.0% darker than its corresponding bulk sample (Exp. #3), which suggests that the larger particles in the bulk sample act to increase the overall albedo of dark soil samples. This indicates that the largest particle size fraction (>1 mm) acts to "normalize" the observed surface brightness relative to the underlying finer fraction, which is a consequence of the formation of the desert pavement lag that is ubiquitous throughout the valleys [77]. Both bulk soil samples (Exps. #1 and #3) also share more similar spectral features than the finer soil fractions, which suggests that in addition to homogenizing the surface brightness, the larger particle sizes are also more compositionally similar than the finer fraction of soils, consistent with previous studies that describe the formation, development, and evolution of desert pavements throughout the MDVs [35,71,72].

*4.2. Relating Surface Albedo and Gravimetric Water Content*

The relationships between measured GWC and sample albedo (relative to air dried samples) for all four experiments are shown in Figure 6. The data can be divided into five regions based on their spectral behavior relative to the measured GWC:

- Region #1. The uppermost optical surface of the sample is dry, with the recorded albedo equivalent to that of the air-dried samples. Because the depth of each soil experiment was held constant, the dominant control on the width of Region #1 is the capillarity of the soil, which influences the ability of water to rise through the sample.
- Region #2. The surface albedo linearly decreases with increasing GWC. The slope and range of the GWC values observed in Region #2 are variable and are primarily related to the dominant surface particle size, which controls both the matric potential and permeability of the soils [40,69].
- Region #3. The local albedo minimum marks the maximum GWC observable using our empirical technique. For the reasons discussed below, we consider this minimum to indicate that the surface is "optically saturated".
- Region #4. A localized increase in apparent albedo with increasing GWC is observed in all samples and is associated with the transition from full saturation (lower GWC boundary), through partial inundation (albedo peak), and then to full inundation (upper GWC boundary). This feature is due to specular reflection (i.e., glinting) from individual surface grains as the surface tension of the water causes distortion around the grains. Region #4 is clearly resolvable when particle sizes are small (Exps. #2 and #4), but much more complex when particle sizes become large and heterogeneously wet during the experiment.
- Region #5. Once the samples are fully inundated, the albedo of the sample will continue to slowly decrease as water abundance (depth) increases.

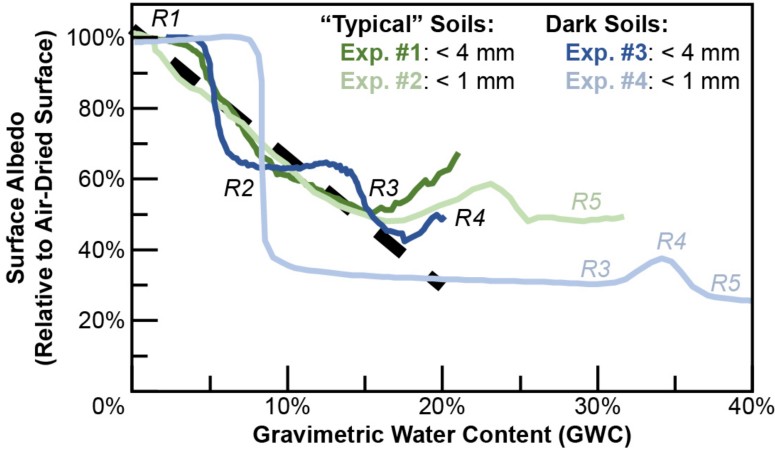

**Figure 6.** The relationship between gravimetric water content and relative surface albedo for each of the four soil samples studied in this investigation. The different spectral regions (R#) are noted. The thick black dashed line represents the best-fit linear relationship between these two variables for all four samples through Region #2.

Exps. #1 and #2 show similar trends between relative albedo and GWC, particularly in Region #2 (the transition from dry to damp to "optically saturated" conditions). One clear difference is that Exp. #1 appears to reach its peak albedo with ~2% GWC remaining in the sample, while the albedo of Exp. #2 does not reach its peak until GWC values much closer to 0%. The observed trend between GWC and surface albedo for Exp. #3 roughly shares the same slope between 0% and 20% GWC values as Exps. #1 and #2, but the observed relationship shows significant differences relative to these other experiments. Between GWC values of 5% and 10%, Exp. #3 demonstrates a lower albedo than either Exps. #1 or #2. Alternatively, Exp. #3 demonstrates a higher albedo than either Exps. #1 or #2 at

GWC values between 10% and 15%. Despite these differences, Exps. #1 and #3 show nearly identical patterns in albedo at <5% GWC, indicating that the final stages of surface desiccation are comparable for both <4 mm particle size samples. The minimum albedo observed in Exps. #1 through #3 fall consistently between 40% and 50% that of their air-dried counterparts, while the associated soil moisture contents at these minimum observed albedos fall between 15% and 18% GWC.

The association between the GWC and observed albedo in Exp. #4 is unique relative to the other soil samples. Exp. #4 does not appear to darken until GWC values > 8%, at which point the observed albedo decreases dramatically to a relative albedo of < 40% over a range of less than 2% GWC. Exp. #4 also shows a relatively stable albedo (between 30% and 40%) over the GWC range from ~10% to ~30%, which is unique to Exp. #4 and does not appear in the three other experiments.

These experiments demonstrate how the dominant particle size of the surface material most significantly influences Region #2 (Figure 6). Comparing Exps. #2 and #4 (the two finest fractions of soil with different dominant particle size distributions) shows how both the slope and range of GWC values observed in Region #2 can vary significantly, with the fine sand-dominated Exp. #4 showing more dramatic changes in albedo over smaller ranges of GWC than the coarse sand-dominated Exp. #2. Interestingly, the slope and GWC range of the coarse sand-dominated Exp. #2 closely matches that of Exp. #1 and the general slope/trends observed in Exp. #3. This suggests that unconsolidated surfaces dominated by coarse sand particles or larger follow similar relationships between observed relative surface albedo and soil moisture content.

The shape of Region #2 in Exp. #3 is interpreted to be a combination of the patterns observed in Exps. #1 and #4. Region #2 in Exp. #3 does not begin until ~5% GWC, when the relative albedo dramatically decreases at roughly the same slope as that observed in Exp. #4, suggesting that the initial wetting of the underlying fine sands control this aspect of the curve. Once these finest grains are all damp, but there is not enough free water available for wicking to wet the largest surface grains, the relative albedo of the sample is able to stabilize even as GWC continues to increase between 6.5% and 13.5% GWC. Beyond GWC values of ~13.5%, water is able to wick up the sides of the larger granules and pebbles to darken these surfaces, resulting in another decrease in relative albedo until optical saturation is observed at a GWC of ~18%, at which point Exp. #3 is essentially indistinguishable from Exps. #1 and #2.

All of these experiments indicate that most surfaces throughout the Fryxell basin, whose uppermost ~5 cm are almost entirely dominated by particle sizes ranging from coarse sands to pebbles and cobbles in a variety of combinations [35,36,78], would share the same relationship between albedo and soil moisture as those demonstrated in the coarse sand- to pebble-dominated trends observed here (Exps. #1 through #3). As demonstrated in Figure 6, relatively small inherent variations in dry surface albedo have less of an effect on the relationship between GWC and relative surface albedo than surface particle size. Fitting a trend in data from Exps. #1 through #3 between relative albedo and GWC values of 0% and 20% yields the following relationship between these two variables:

$$GWC = -0.271\alpha + 0.284 \tag{2}$$

where $\alpha$ is the measured surface albedo relative to the long-term baseline (dry) albedo ($r^2$ = 0.863, average error = $\pm1.6\%$).

### 4.3. Application to Remote Sensing Data

The above equation was applied to the 21 highest quality WV02 and WV03 relative albedo products (derived as described above) to calculate the GWC at each pixel at specific time points when the images were acquired. While individual images provide a glimpse into the region's instantaneous hydrologic regime, products that effectively compare images to each other and identify broad spatial and temporal patterns are most relevant to hydrological and ecological processes.

The basin-wide results of our investigation can be found in Figure 7. Figure 8 focuses on a small region at the base of Canada Glacier and a few hundred meters west of Bowles Creek and Green Creek. Towards the top of this scene is a glacial melt pond that is regularly fed directly from Canada Glacier meltwater. A region of enhanced soil moisture is evident at the bottom of this image; this is the terminal reaches of a water track that originates along the slopes of Nussbaum Riegel to the southwest. Both the melt pond and the water track appear as regions of enhanced soil moisture (Figure 8b), with the average GWC for the melt pond, the water track, and the surrounding landscape modeled at 6.7 ± 2.5%, 4.4 ± 1.2%, and 1.7 ± 0.3%, respectively. Figure 8c shows the fraction of satellite images collected between 2009 and 2019 when the soils are modeled as >6.1% GWC (the estimated soil wilting point of the finer soil sample). The small melt pond exceeds the wilting point in approximately 40% of the 21 images used in this study, while the water track exceeds the wilting point in approximately 20% of the images. The surrounding soils rarely exceed 6.1% GWC, which is consistent with previous studies and field observations. Finally, Figure 8d shows the fraction of satellite images where the soils are modeled as >12.3% GWC (the measured field capacity of the finer soil sample). Neither the typical interfluve soils nor the water tracks are ever observed to exceed field capacity, but portions of the melt pond are shown to exceed field capacity in roughly half of the 21 images used in this study. This example demonstrates how our techniques are able to easily identify soils with unique hydrological settings.

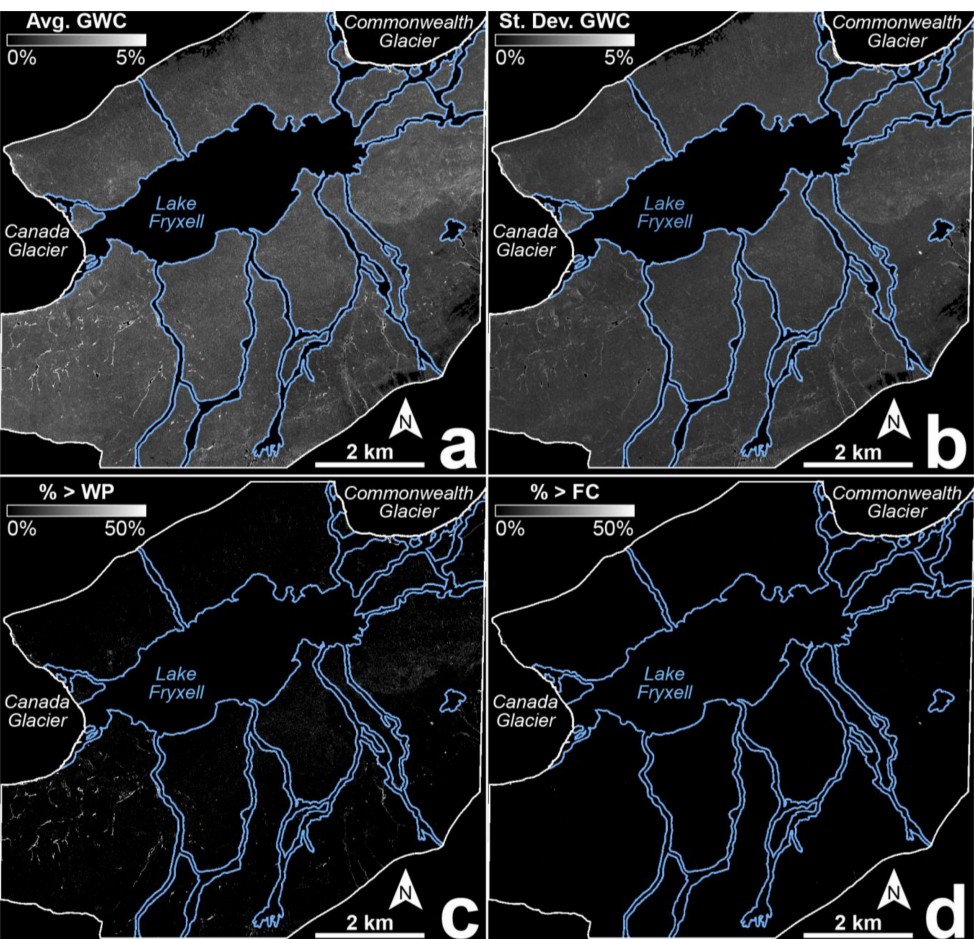

**Figure 7.** (**a**) Average soil gravimetric water content (GWC) calculated using 21 individual satellite images acquired since 2009; (**b**) the standard deviation of GWC calculated using the same images as (**a**); (**c**) the frequency that a given pixel is observed to exceed the soil's estimated wilting point (WP; 6.1% GWC); and (**d**) the frequency that a given pixel is observed to exceed the soil's measured field capacity (FC; 12.3% GWC). Source imagery © 2009–2021 Maxar.

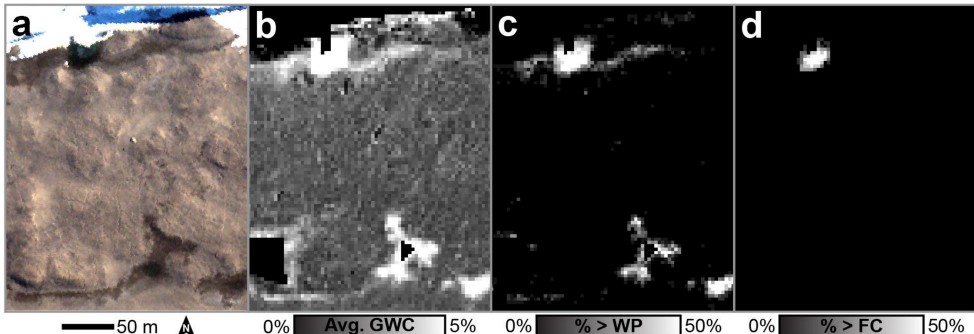

**Figure 8.** (**a**) True color image of an area south of Canada Glacier, which is seen at the upper edge of the image; (**b**) average gravimetric water content (GWC) of the soil in the 21-image composite developed in this study; (**c**) the fraction of the time each pixel is observed to exceed the estimated wilting point (WP) of 6.1% GWC; and (**d**) the fraction of the time each pixel is observed to exceed the measured field capacity (FC) of 12.3% GWC. Source imagery © 2009–2021 Maxar.

Targeted temporal studies are also possible using this generated GWC dataset. For example, Figure 9 shows a region near the upper reaches of Harnish Creek in the Fryxell basin where an incised channel transitions to a depositional regime before reaching Lake Fryxell. Three GWC products acquired over the course of one month in January–February 2017 are also shown, with their topographically removed surface albedo products shown in Figure 9b–d. Transects (shown as a dashed yellow line in Figure 9a) through this fluvial feature are shown below the images in Figure 9 and clearly show how this fluvial system was drying between late January and late February. Visually, the soils appear to brighten between these three timesteps, with the final image actually showing a brighter channel and sediment than the surrounding landscape, indicating either the presence of frozen water ice or salts. The yellow star in all images is also shown in the transect plot below, confirming that the soils are drying over time. The recorded GWC values are also consistent with field observations; a GWC between 5 and 10% throughout the channel in Figure 9b is suggestive of damp, yet unsaturated soils.

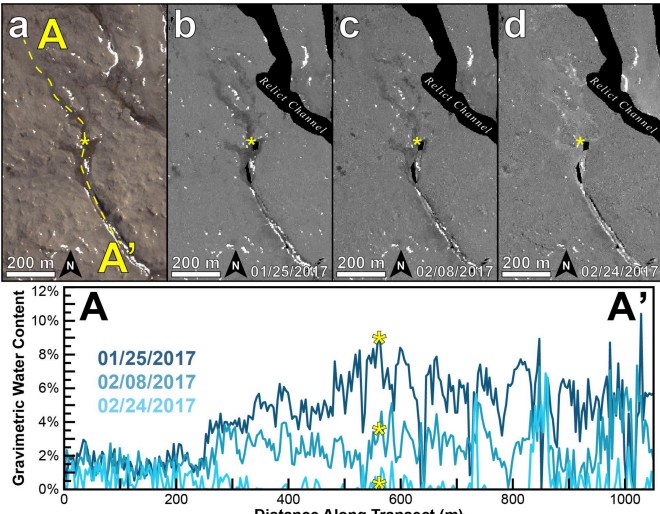

**Figure 9.** (**a**) True color image of a region south of Lake Fryxell, Taylor Valley, Antarctica, that demonstrates observed changes in soil moisture throughout the season; (**b**–**d**) show derived gravimetric water content (GWC) maps for the same regions scaled in an identical manner, showing a brightening over time. The chart at the bottom shows the evolution of these GWC transects over time, with the final date showing little, if any, soil moisture remaining. The location of the yellow stars in (**a**–**d**) are represented in the profiles in the bottom chart. Imagery © 2017 Maxar.

## 5. Discussion

Our work provides a new technique for accurately estimating surface soil moisture content using orbital remote sensing data, which can be applied to data collected both in the past and in the future. These methods are insensitive to solar illumination geometry as our lidar-based topographic removal technique normalizes the surface albedo in response to varying solar illumination. The technique is also able to derive accurate results even in the presence of snow or salts throughout the landscape. As such, we have demonstrated that our technique is capable of rapidly and accurately estimating surface soil moisture abundance and distribution throughout the Fryxell basin.

Ref. [53] observed a median surface darkening of 27.7% in MDV soils during high humidity events, suggesting a significant flux of water vapor from the atmosphere into the soils. They compared the observed darkening to the work of [21] and suggested that the observed darkening corresponds to a GWC of roughly 2–3% [53]. However, our model derived above predicts a GWC of ~9% when soil albedo decreases by this magnitude (27.7%), suggesting that the soil moisture measurements of [53] could be an underestimate by a factor of roughly three and represent an ecologically meaningful soil water content. This also suggests that the process of deliquescence in high-humidity environments can result in soils that approach the GWC threshold identified by [15], where nearly 100% of soil nematodes uncoil and resume physiological and metabolic functions in just six hours. Both the magnitude and the duration of the soil excursions suggested by [53] suggest that the soils of Taylor Valley have the potential to host active ecosystems whose populations experience significant durations of habitable (and potentially even clement) environmental conditions.

Unlike more temperate desert environments on Earth, the MDVs host significant reservoirs of water stored as glacial ice, shallow subsurface ice (i.e., permafrost), and surficial snow deposits. Climate models predict significant warming over East Antarctica through the year 2100 at a projected rate of +0.03 °C yr$^{-1}$ [79], despite the modest cooling observed in the MDVs in the 1990s and early 2000s [80]. Ref. [47] demonstrates how some of these reservoirs are only metastable under current Antarctic environmental conditions, not only due to rising temperatures, but also due to the complex interplay between soil thermophysics and sediment/albedo feedback, resulting in insolation-driven melting. In combination with rising temperatures, it is hypothesized that both the hydrology and ecology of the MDVs will likely respond to a growing amount of landscape instability [46]. For example, the loss of subsurface permafrost in the Fryxell basin could lead to landscape deflation and melt channel migration, putting established ecosystems at risk. Similarly, even minor changes to subsurface ice volumes can result in the redirection of water tracks throughout the landscape, migrating water and dissolved ions away from previously established routes [17]. Finally, in the MDVs, where summer temperatures hover near 0 °C, additional warming will also result in a greater likelihood of snow/ice melt and the greater possibility of rainfall. Unlike any other process, rainfall has the ability to increase soil moisture ubiquitously across the landscape regardless of proximity to other sources of water.

Our data can be used to help refine models aimed at investigating the composition, structure, and functions of ecosystems throughout the MDVs and to make better predictions about ecosystem responses to present and future environmental change. The habitat suitability of Antarctic soil biota has been well documented throughout the MDVs. For example, "large" endemic terrestrial invertebrates such as Collembola and Acari are known to only persist in areas with access to high soil moisture abundances [81,82]. Tardigrades, rotifers, and the nematode species *Plectus antarcticus* and *Eudorylaimus antarcticus* also show positive correlations between soil moisture and population abundances, while populations of the nematode species *Scottnema lindsayae* show a negative correlation with variation in soil water [83].

The observed relationship between measured GWC and the relative albedo of the soils is primarily dependent on particle size distribution, which determines both the amount of

water that can be held within the soil [84] as well as the migration of water through the soils towards the surface as evaporation progresses [85]. While we attempted to minimize the effects of the water migration rate on the experiment by keeping our prepared samples as shallow as possible (maximizing the connectivity between the surface and the subsurface), the disconnect between surface moisture (as indicated by observed darkening) and GWC (as indicated by the sample weight during the experiments) is still apparent in both the rate of albedo change as well as the recorded GWC when the surface is observed to be completely dry. These observations suggest two major hindrances for understanding soil moisture content in MDV soils. First, our ability to observe and quantify soil moisture content remotely will always be an underestimate of the actual water content of the soils, as there is no accurate way of estimating subsurface soil moisture in the MDVs using optical remote sensing at present (Table 2). Neither the active layer depth nor the amount of soil moisture in the subsurface can be accurately estimated using our technique, and so additional investigation must be performed to refine these estimates or to understand the mechanisms underlying the variation in surface expression of depth-integrated water content. Second, while the uppermost surface of well-developed soils and active layers in the MDVs are known to be dominated by coarse sands, granules, pebbles, and cobbles, the particle size distribution, extent of sorting, and cohesion of the subsurface soil can have significant influences on the rate and nature of capillarity, matric potential, and water vapor movement upwards through the soil column. Figures 4 and 6 and Table 2 demonstrate how two distinct soils from close proximity within the Fryxell basin can have very different bulk soil properties (i.e., composition, particle size distribution). However, the dominance of coarse particles at the uppermost surface results in similar relationships between the GWC and observed relative albedo. The variability of surface (and subsurface) soil properties throughout the Fryxell basin means that additional field work will be needed to characterize the full extent of this diversity and its influence on observed spectral properties. Nonetheless, the frequency and widespread distribution of wet soils outside of channelized streams throughout the Fryxell basin demonstrates the importance of shallow subsurface flow, snowmelt, percolation through the soil, and other previously underappreciated modes of hydrologic activity.

**Table 2.** Estimated volumes of liquid water present throughout the Fryxell basin based on the average gravimetric water content calculated for each pixel and the lab results discussed here. The estimated volume of Lake Fryxell ($4.3 \times 10^7$ m$^3$) was derived from [86].

| Depth of Moist Active Layer | Interfluve Liquid Water Volume (Fryxell Basin) | Water Equivalent Layer (Fryxell Basin) | Fraction of Mean Annual Precipitation (50 mm weq [28]) |
|---|---|---|---|
| 0 cm | 0 m$^3$ | 0 mm | 0% |
| 0.1 cm | $220 \pm 73$ m$^3$ | $0.02 \pm 0.01$ mm | 0.04% |
| 1 cm | $2197 \pm 735$ m$^3$ | $0.15 \pm 0.05$ mm | 0.30% |
| 5 cm | $10,984 \pm 3674$ m$^3$ | $0.77 \pm 0.26$ mm | 1.54% |
| 25 cm | $54,918 \pm 18,370$ m$^3$ | $3.84 \pm 1.28$ mm | 7.68% |
| 50 cm | $109,836 \pm 36,741$ m$^3$ | $7.67 \pm 2.57$ mm | 15.34% |
| 100 cm | $219,673 \pm 73,482$ m$^3$ | $15.35 \pm 5.13$ mm | 30.70% |

## 6. Conclusions

Our work has demonstrated the ability to derive surface soil GWC using remote sensing data, long-term baseline surface albedos, a topographic correction technique to minimize the influences of viewing and illumination geometry, and lab experiments to quantitatively relate observed changes in surface albedo to volumetric changes in liquid water. We show that the interfluve regions of the Fryxell basin of Taylor Valley are hydrologically dynamic regions that can frequently reach soil moisture contents capable of

supporting regional ecosystem processes, supporting the remote identification of photoautotrophic communities throughout the landscape [23]. In tandem, these results suggest a complex relationship between the distribution of photoautotrophic communities and hydrological connectivity throughout the landscape.

The importance of liquid water for sustaining ecosystem processes in the MDVs makes the study of its distribution and availability over space and time all the more critically important. As extensively described in previous literature, the soil moisture throughout the MDVs is highly variable and patchy in nature as a result of the spatial heterogeneity in the depth of buried ice, proximity to other sources of water, and other factors including the presence of salts and the subsurface ice table topography. The ability to use high-resolution remote sensing data to assess the distribution and abundance of surface soil moisture present throughout space and time is a critically valuable tool for understanding ecosystem functioning. Future ground validation can also help to refine our models and to correlate ecological properties with the instantaneous and average decadal water availability maps produced in this study. Lastly, the continued characterization of soil moisture availability using remote sensing data over the coming years will provide critical information about how the oft-regarded "stable" Antarctic landscape and the unique MDV ecosystem are rapidly responding to an ever-changing climate.

**Supplementary Materials:** The following supporting information can be downloaded at: https://www.mdpi.com/article/10.3390/rs15123170/s1, and includes all of the spectral and soil moisture information related to the lab experiments performed in this investigation. Data are also archived with the Environmental Data Initiative (EDI).

**Author Contributions:** M.R.S., J.E.B., E.R.S., J.S.L. and M.N.G. conceptualized this project and contributed to acquiring funding. M.R.S., L.E.F., S.N.P., H.M.M. and B.S. contributed to laboratory analyses. M.R.S., J.E.B., E.R.S., J.S.L., L.C.K., M.N.G., B.J.A., J.P.K. and P.T.D. all contributed substantially to contextualizing our results with other scientific results. M.R.S. led the preparation of the original manuscript draft, while all authors contributed equally to the revision and editing of subsequent versions of this manuscript. All authors have read and agreed to the published version of the manuscript.

**Funding:** This research was funded by the US National Science Foundation (NSF) Office of Polar Programs, Award No. 2046260 to M.R.S. and Award No. 2045880 to E.R.S.

**Data Availability Statement:** Multispectral commercial data are freely available to NSF-funded investigators through a cooperative agreement with the National Geospatial Intelligence Agency (NGA). Data outside of the scope of this cooperative agreement are available for purchase from Maxar, Inc. Hyperspectral data collected for this investigation are provided as Supplementary Materials.

**Acknowledgments:** The authors would like to acknowledge and thank the US National Science Foundation, the Long Term Ecological Research Program, the US Antarctic Program (and all affiliated contractors and field personnel), helicopter pilots and technicians, and others who help to make Antarctic field work possible.

**Conflicts of Interest:** The authors declare no conflict of interest.

## Appendix A

**Table A1.** List of WorldView-2 and -3 data used in this study.

| Platform | CAT/Image ID | Time of Acquisition (GMT) | Part of High-Quality Subset? |
|----------|--------------|---------------------------|------------------------------|
| WV02 | 1030010003085D00 | 12/25/2009 18:40:39 | Yes |
| WV02 | 1030010003085D00 | 12/25/2009 18:40:40 | |
| WV02 | 1030010003AF300 | 12/25/2009 21:59:36 | Yes |

**Table A1.** *Cont.*

| Platform | CAT/Image ID | Time of Acquisition (GMT) | Part of High-Quality Subset? |
|---|---|---|---|
| WV02 | 1030010003A9F300 | 12/25/2009 21:59:37 | |
| WV02 | 1030010003921200 | 01/10/2010 20:36:08 | |
| WV02 | 1030010004306300 | 02/13/2010 21:36:33 | Yes |
| WV02 | 1030010007B79300 | 10/24/2010 19:48:32 | Yes |
| WV02 | 103001000834C800 | 11/20/2010 21:47:15 | |
| WV02 | 1030010008188100 | 11/27/2010 20:53:06 | |
| WV02 | 103001000823BF00 | 12/16/2010 21:04:04 | |
| WV02 | 103001000823BF00 | 12/16/2010 21:04:05 | |
| WV02 | 1030010008A27200 | 12/19/2010 20:55:05 | |
| WV02 | 1030010008A27200 | 12/19/2010 20:55:06 | Yes |
| WV02 | 103001000825E900 | 12/22/2010 20:46:12 | |
| WV02 | 1030010007823D00 | 12/25/2010 15:39:57 | Yes |
| WV02 | 103001000823AB00 | 12/26/2010 20:00:17 | |
| WV02 | 1030010009319D00 | 02/09/2011 21:03:38 | |
| WV02 | 1030010009319D00 | 02/09/2011 21:03:40 | |
| WV02 | 103001000FA6A800 | 11/01/2011 17:47:36 | |
| WV02 | 103001000F2F9F00 | 12/23/2011 19:15:40 | |
| WV02 | 10300100102C1200 | 01/20/2012 18:46:02 | |
| WV02 | 103001001485B00 | 01/29/2012 19:54:39 | |
| WV02 | 10300100111D2700 | 02/03/2012 20:10:26 | Yes |
| WV02 | 10300100111D2700 | 02/03/2012 20:10:27 | |
| WV02 | 103001001B656700 | 09/30/2012 21:06:51 | |
| WV02 | 103001001B656700 | 09/30/2012 21:06:52 | |
| WV02 | 103001001D249700 | 10/31/2012 20:23:17 | |
| WV02 | 103001001D249700 | 10/31/2012 20:23:19 | |
| WV02 | 103001001D642400 | 01/05/2012 21:30:30 | Yes |
| WV02 | 103001001D642400 | 01/05/2012 21:30:31 | |
| WV02 | 103001001D0EE000 | 01/05/2013 21:30:52 | |
| WV02 | 103001001D0EE000 | 01/05/2013 21:30:53 | |
| WV02 | 103001001ED2C100 | 01/05/2013 21:31:38 | |
| WV02 | 103001001ED2C100 | 01/05/2013 21:31:40 | |
| WV02 | 103001001D381500 | 01/05/2013 21:31:58 | |
| WV02 | 103001001D381500 | 01/05/2013 21:31:59 | |
| WV02 | 103001002BD0AD00 | 12/27/2013 19:37:53 | |
| WV02 | 103001002CCCEE00 | 02/01/2014 20:52:40 | |
| WV02 | 103001002CCCEE00 | 02/01/2014 20:52:41 | |
| WV02 | 103001003ED2B400 | 01/21/2015 19:51:58 | Yes |
| WV03 | 1040010007721800 | 01/22/2015 20:12:31 | Yes |
| WV02 | 103001003CD3ED00 | 01/23/2015 20:17:49 | Yes |
| WV02 | 103001003D133E00 | 02/16/2015 20:33:10 | |

**Table A1.** *Cont.*

| Platform | CAT/Image ID | Time of Acquisition (GMT) | Part of High-Quality Subset? |
|---|---|---|---|
| WV02 | 103001004CAD8700 | 11/01/2015 20:12:42 | |
| WV03 | 1040010015737400 | 12/01/2015 21:15:44 | Yes |
| WV03 | 1040010018D33600 | 02/17/2016 20:09:42 | |
| WV02 | 10300100648C3100 | 01/25/2017 21:14:28 | Yes |
| WV03 | 104001002855C000 | 01/28/2017 21:00:46 | Yes |
| WV02 | 10300100643A1400 | 02/08/2017 20:57:28 | Yes |
| WV03 | 1040010029308E00 | 02/10/2017 21:09:28 | Yes |
| WV02 | 1030010063108000 | 02/14/2017 20:35:57 | |
| WV03 | 1040010028C25D00 | 02/15/2017 22:28:18 | Yes |
| WV02 | 10300100662D7100 | 02/25/2017 21:07:26 | Yes |
| WV02 | 1030010077755100 | 01/19/2018 19:06:05 | |
| WV02 | 1030010089D13500 | 12/11/2018 21:00:21 | Yes |
| WV03 | 10400100485D6900 | 01/26/2019 21:46:20 | Yes |
| WV02 | 103001009FA0AE00 | 12/03/2019 20:20:55 | Yes |

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
