# Peer review of "The Distribution of Surface Soil Moisture over Space and Time in Eastern Taylor Valley, Antarctica"

_remotesensing, doi:10.3390/rs15123170_

Round 1

Reviewer 1 Report

This manuscript describes the derivation of a simple retrieval to relate satellite-derived visible and near-infrared albedo to soil moisture content for use in the McMurdo Dry Valleys of Antarctica.  The retrieval is a linear regression derived from laboratory experimental data.  Such a retrieval is apparently important to further biological research in soil ecosystems in the area.

The authors admit that the soil moisture retrieved from this method is not accurate, being always underestimated.  There are already satellites that are specifically designed to measure soil moisture such as SMAP.  Why can’t any of these be used to obtain soil moisture in this region?

Also, at Line 620, the authors state that their technique is not sensitive to brightening from snow or salts.  How is this possible?

The authors should address these questions.  These should be minor revisions to their manuscript.

Author Response

Response to Reviewer #1 Comments:

The authors admit that the soil moisture retrieved from this method is not accurate, being always underestimated.  There are already satellites that are specifically designed to measure soil moisture such as SMAP.  Why can’t any of these be used to obtain soil moisture in this region?

We apologize for the misunderstanding.  Like all techniques, the use of VNIR reflectance spectroscopy to derive surface soil moisture estimates has its limitations.  In this instance, the ability to only observe the uppermost optical surface means that the nature of the subsurface is largely unconstrained.  The spatial resolution of SMAP is more than four orders of magnitude lower than that of the WorldView constellation (Colliander et al., 2017), while the penetration depth of the C-band radar (like that employed in the RADARSAT-2 mission) varies enormously as a function of soil moisture itself, ranging from less than 1 mm depth at ~ 35% volumetric water content to more than 10 cm at < 5% volumetric water content (Nolan and Fatland, 2003).  

We appreciate the author’s recommendation to address these other satellites and instruments.  In doing so, we have bolstered our justification of the use of the WorldView constellation of satellites and the development of this new technique for use in the McMurdo Dry Valleys over the use of other satellite options, as the limitations and uncertainties on reflectance spectroscopy are much better known than those that require unfounded assumptions about subsurface properties.

Also, at Line 620, the authors state that their technique is not sensitive to brightening from snow or salts.  How is this possible?

We have clarified this statement in the manuscript to avoid any confusion.  What we originally meant to say here was that this technique for determining the gravimetric water content of soils can correctly identify when pixels brighten (as a result of the addition of snow or salts) and, therefore, correctly determine that there was not an increase in soil moisture.

Reviewer 2 Report

The manuscript explained the distribution of surface soil moisture over space and time in eastern Taylor Valley, Antarctica using multisepctral satellite remote sensing imagery. Quantification of the soil moisture content was shown throughout the Fryxell basin in the manuscript. The authors believe that this technology is conducive to identify ecological hotspots in the harsh polar Antarctic environment and serve as a baseline for detecting future changes in the soil hydrological regime. A large amount of WorldView satellite remote sensing data is being used. This manuscript has a certain degree of innovation. But there were still some doubts that need to be clarified:

(1) The manuscript needs to be carefully proofread. e.g. There is an extra "." in "3.2. . Quantitatively Relating Relative Albedo to Gravimetric Water Content".  

(2) What do the "WP" and "FC" represent in Figure 4 and Figure 8. It's not clear.

(3) Line 677: What does the underlined "underestimate" represent?

(4) It's best to indicate the sampling "two sets of soils from the Fryxell basin" locaiton in the Figure .

(5) The boundary between Figure 8c and Figure 8d is unclear?

All in all, The theme of the manuscript is very good. It quantitativley relates observed changes in surface albedo to volumetric changes in liquid water based on Lab experiments. However there is too less description in the conclusions. I hope the authors can expound it.

Author Response

Response to Reviewer #2 Comments:

(1) The manuscript needs to be carefully proofread. E.g., There is an extra “.” in “3.2. . Quantitatively Relating Relative Albedo to Gravimetric Water Content”.

We have carefully proofread the manuscript again prior to resubmission.  Thank you for your careful edits.

(2) What do the “WP” and “FC” represent in Figure 4 and Figure 8. It’s not clear.

WP and FC refer to the wilting point and field capacity of the soils, respectively.  Figure 4 does not reference WP or FC, and we have defined these acronyms where explained in the figure captions of Figures 7 and 8.  Thank you for the recommendation.

(3) Line 677: What does the underlined “underestimate” represent?

We apologize for the formatting error.  We have removed the underline.

(4) It’s best to indicate the sampling “two sets of soils from the Fryxell basin” location in the Figure.

Thank you for this recommendation.  We have added the locations of our two soil samples to Figure 1.

(5) The boundary between Figure 8c and Figure 8d is unclear?

Thank you for identifying this.  We have changed these borders to white to make them more differentiable.

Too little description in the conclusions, should be expounded upon.

Thank you for this recommendation.  We have added an additional paragraph that highlights our quantitative results before discussing the broader conclusions and implications of our work.

Reviewer 3 Report

The introduction is too long

Author Response

Response to Reviewer #3 Comments:

We apologize for the seemingly long introduction.  We feel that the introduction, especially with the edits recommended by the other reviewers, appropriately details the need for understanding the spatial and temporal variability of soil moisture in the McMurdo Dry Valleys.  We tried to move some of this material to the Background section, but found that doing so hid much of the important motivating information regarding nematode activity, etc.  Therefore, we feel that our five-paragraph introduction seems appropriate given the scope of the impact that our work has on the field.